# Stair-Climbing Training with Interferential Electrotherapy Improves Knee Muscle Strength, Dynamic Postural Stability, Pain Score, and Physical Activity in Patients with Knee Osteoarthritis

**DOI:** 10.3390/diagnostics14182060

**Published:** 2024-09-17

**Authors:** Jin Hyuck Lee, Gyu Bin Lee, Woo Yong Chung, Ji Won Wang, Ki-Mo Jang

**Affiliations:** 1Department of Sports Medical Center, Anam Hospital, Korea University College of Medicine, Seoul 02841, Republic of Korea; gnkfccc@hanmail.net (J.H.L.); humanwellness@naver.com (G.B.L.); jwy8098@naver.com (W.Y.C.); jiwon-eee@naver.com (J.W.W.); 2Department of Orthopaedic Surgery, Anam Hospital, Korea University College of Medicine, Seoul 02841, Republic of Korea

**Keywords:** knee osteoarthritis, knee muscle strength, dynamic postural stability, pain, physical activity, stair-climbing training

## Abstract

**Background/Objective:** This study aimed to compare the functional outcomes, such as knee muscle strength, dynamic postural stability, pain scores, and physical activity, in patients with knee osteoarthritis (OA) on stair climbing training with and without interferential electrotherapy (IFE) for 12 weeks. **Methods:** A total of 40 knee OA patients with Kellgren–Lawrence (K–L) grade ≤ 2 were enrolled (20 stair-climbing training with IFE vs. 20 stair-climbing training without IFE). The knee quadriceps and hamstring muscle strengths were measured using an isokinetic device. The dynamic postural stability was assessed using postural stabilometry. The pain score was evaluated using the visual analog scale (VAS). Physical activity was assessed using the Western Ontario and McMaster Universities Osteoarthritis Index (WOMAC). **Results:** The WOMAC score was significantly different (*p <* 0.019) between stair-climbing training with and without IFE in patients with knee OA, while knee muscle strength, dynamic postural stability, or pain score were not (all *p >* 0.05). **Conclusion:** Stair-climbing training with IFE was more beneficial for physical activity recovery than stair-climbing training without IFE. Therefore, clinicians and therapists should be aware that stair climbing, which can be practiced in daily life for the management of patients with knee OA, and the addition of IFE may improve physical activity.

## 1. Introduction

Knee osteoarthritis (OA) is a common musculoskeletal disorder in individuals over 60 years of age [1] that causes pain, swelling, and stiffness and reduces quality of life [2,3]. In particular, patients with knee OA have reduced physical activity, such as walking, squatting, and stair climbing, than those without knee OA [4,5]. Therefore, therapeutic exercise as a rehabilitation treatment is important to reduce pain and improve physical activity in patients with knee OA.

Patients with knee OA experience knee muscle weakness and proprioception loss [6,7,8,9], which causes joint instability and joint stress, thus increasing pain and accelerating knee OA progression. Previous studies have reported the benefits of surgery [10,11] and glucocorticoid injection [12,13] for knee OA management in improving knee muscle strength, postural stability, and pain scores. However, a Cochrane review [14] and recent studies [15,16] have recommended therapeutic exercises, such as muscle strengthening and proprioceptive exercises, to improve pain scores and physical activity in patients with knee OA [12,16,17]. In particular, therapeutic exercises have greater benefits than glucocorticoid injections for physical activity [10,12]. Among the therapeutic exercises, stair-climbing training was added to the rehabilitation protocol for knee muscle strength and proprioception recovery in patients with knee injuries [18,19,20]. However, patients with knee OA may experience adverse effects due to high joint pressure during stair climbing [17,21]. To date, no studies have investigated whether stair-climbing training has adverse effects on functional outcomes such as knee muscle strength, postural stability, pain scores, and physical activity in patients with knee OA. Furthermore, in our clinical experience and in many other institutions, electrotherapy is often used to reduce pain, increase range of motion, and stimulate muscles in patients with knee OA. However, whether electrotherapy, commonly used for pain control in patients with knee OA [22], can improve functional outcomes has not been clarified yet.

This study aimed to compare functional outcomes, including knee muscle strength, postural stability, pain scores, and physical activity, in patients with knee OA undergoing stair-climbing training with and without interferential electrotherapy (IFE) over a 12-week period. We hypothesized that stair-climbing training with IFE would improve all parameters in patients with knee OA than stair-climbing training without IFE.

## 2. Methods

### 2.1. Study Design and Patient Enrollment

This prospective comparative study followed CONSORT guidelines for non-pharmacological treatments and compared functional outcomes including knee muscle strength, postural stability, pain scores, and physical activity in patients with knee OA undergoing stair-climbing training with and without IFE over a 12-week period. Sixty knee OA patients over 60 years old, with unilateral Kellgren–Lawrence (K–L) grade ≤ 2, who were diagnosed by an orthopedic surgeon and had continuous pain for at least 3 months from January 2021 to January 2023, were recruited. We excluded 20 patients for the following reasons: lateral tibiofemoral narrowing, chondromalacia, prior knee surgery and injections in the past 6 months, use of oral corticosteroids within 4 weeks, inability to walk without a walking aid, neurological conditions such as spine and vestibular injuries, and increased pain during the test. Patients with bilateral knee OA were excluded from this study because they were at higher risk of falls due to higher joint pressure during stair-climbing training [23,24]. Of the total 60 patients with knee OA, 20 were excluded, leaving 40 patients who were then allocated into two groups (Figure 1). All patients provided written informed consent prior to participating in this study, which was approved by the institutional review board (No. 2021AN0031).

### 2.2. Interventions

#### 2.2.1. Stair-Climbing Training

Participants assigned to the stair-climbing training group underwent a gradual stair-climbing training program three times a week for 12 weeks. The total number of stairs on the sixth floor of our institution was 144, dividing this into three sets of two floors (2 floors × 3 sets) for each training session (the width, depth, and height of one stair were 126.5, 28.5, and 16.5 cm, respectively; Figure 2A), with 10–15 min rest periods between sets. This was repeated two, three, four, and five times for weeks 1–3, 4–6, 7–9, and 10–12, respectively. If the visual analog scale (VAS) score was ≥5 and persisted during stair-climbing training, the training was stopped and re-started when the pain disappeared. However, stair descent was not performed during stair-climbing training because of the adverse effects of increased load on the knee joint [25], and an elevator was used.

#### 2.2.2. Interferential Electrotherapy (IFE)

Participants assigned to the interferential electrotherapy group underwent interferential current (IFC) therapy (Figure 2B) with stair-climbing training for 15 min, three times a week for 12 weeks. IFC was applied after stair-climbing training at an acceptable threshold intensity, using four electrodes positioned around the knee. The beat, sweep frequencies, and amplitude-modulated frequencies were 100, 150 Hz, and 100 Hz, respectively [22,26,27].

### 2.3. Outcome Measurement

#### 2.3.1. Knee Muscle Strength Assessment

Knee muscle strength was measured using Biodex Multi-Joint System 4 (Biodex Medical Systems, Shirley, NY, USA) [28]. All patients were aligned with the rotational axis of the isokinetic machine and the lateral femoral condyle in a seated position with the hips and knees flexed at 90° each on a chair. Before testing, the participants performed three submaximal repetitions at 60°/s of flexion and extension, followed by five maximal contractions. Gravity correction for torque values using the Biodex Advantage Software v5.3 was performed at 30° knee extension.

Knee muscle strength was measured in affected knees for the quadriceps and hamstring muscles by maximum torque normalized to the body weight (peak torque/body weight, N·m/kg × 100). A previous study revealed a good intraclass correlation coefficient (ICC) of isokinetic peak torque in patients with knee OA (0.83 and 0.89 for the quadriceps and hamstring muscles, respectively) [24].

#### 2.3.2. Postural Stability Assessment

Postural stability was assessed for dynamic balance ability using the Biodex Stability System (BSS), which employs a dynamic foot platform that tilts from 0° to 20° in any direction and allows for 360° rotation, which is used to evaluate proprioception [7,29]. The test was performed in 2 trials, each lasting 20 s, with a rest period of 10 s between trials. In this study, the tilt angles (in degrees) in the anterior–posterior (Y) and medial–lateral (X) directions were sampled by the BBS software v4.0 system and converted to overall stability index (OSI) values, which were used as the stability index; OSI $=\sqrt{\frac{\sum {\left(0-x\right)}^{2}+\sum {\left(0-Y\right)}^{2}}{number\;of\;samples}}$.

In a previous study, the ICCs of the postural stability test for individuals over 60 years of age were acceptable for OSI (ICCs = 0.69) [30], which is an important indicator of postural stability [31]. For safety reasons, all patients were assessed for six levels of dynamic postural stability: standing comfortably on both legs, with both hands on the hips, and with eyes open. Level 12 is the most stable, whereas level 1 is the most unstable. A low stability index indicated good postural stability.

### 2.4. Pain Assessment

The VAS was used to assess pain levels in patients with knee OA [21], where 0 indicated no pain and 10 indicated the worst pain. Based on a previous study in patients with knee OA, two point difference in the VAS score after intervention was defined as the minimal clinically important difference (MCID) [32].

#### 2.4.1. Questionnaire

The Western Ontario and McMaster Universities Osteoarthritis Index (WOMAC) was used to evaluate pain and physical activity in patients with knee OA [33]. WOMAC consists of 24 questions, including pain (five items), stiffness (two items), and function (17 items), with high scores indicating limitations in function and symptom severity. As reported in a previous study [34], the MCID for the WOMAC was 4–6.8 points.

#### 2.4.2. Sample Size Estimation and Statistical Analysis

A priori power analysis was performed to determine the sample size at an alpha level of 0.05 and a power of 0.8. Based on previous reports, an OSI difference >0.3 between groups with knee OA [35] and a quadriceps strength difference >15% between knees with and without pain [36] were regarded as clinically important. In the present study, 38 patients were required to detect a significant difference between postural stability and quadriceps strength in the two groups, and the power to detect between-group differences in postural stability and quadriceps strength was 0.805 and 0.789, respectively.

The Shapiro–Wilk test was used to assess the data distribution normality. The chi-square test was used to compare differences in demographic characteristics between groups.

A paired *t*-test was used to compare differences in knee muscle strength, postural stability, pain scores, and physical activity before and after 12 weeks of interventions in each group. Student’s *t*-test and chi-square test were used to compare variables between the two groups. Data were analyzed using SPSS 22.0, and *p <* 0.05 was considered statistically significant.

## 3. Results

The baseline demographic characteristics, including sex, affected side, age, height, and weight, were not significantly different between the 40 patients with knee OA enrolled in this study (*p* > 0.05, Table 1).

### 3.1. Comparison of Pre-Test versus Post-Test Outcomes in Each Group

In stair-climbing training with the IFE group, knee muscle strength for quadriceps and hamstring muscles, OSI, VAS score, and WOMAC score significantly improved in the affected knees after 12 weeks of intervention (quadriceps strength [N·m/kg × 100], 61.4 ± 10.5 vs. 90.1 ± 17.2, *p <* 0.001; hamstring strength [N·m/kg × 100], 41.6 ± 14.4 vs. 66.6 ± 17.1, *p <* 0.001; OSI [degrees], 5.6 ± 1.0 vs. 3.1 ± 0.8, *p* < 0.001; median VAS score [point], 5.0 vs. 2.5, *p <* 0.001; median WOMAC score [point], 68.0 vs. 35.0, *p* < 0.001; Figure 3A).

In stair-climbing training without the IFE group, knee muscle strength for quadriceps and hamstring muscles, OSI, VAS score, and WOMAC score significantly improved in the affected knees after 12 weeks of intervention (quadriceps strength [N·m/kg × 100], 59.8 ± 12.5 vs. 76.7 ± 17.1, *p =* 0.031; hamstring strength [N·m/kg × 100], 44.0 ± 15.8 vs. 61.4 ± 19.6, *p =* 0.022; OSI [degrees], 5.2 ± 1.3 vs. 3.3 ± 1.0, *p <* 0.001; median VAS score [point], 5.0 vs. 3.5, *p <* 0.001; median WOMAC score [point], 70.0 vs. 46.0, *p =* 0.008; Figure 3B).

### 3.2. Comparison of Outcome Measures between Groups

Median WOMAC score (point: 35.0 vs. 46.0, *p <* 0.019) was significantly different in the affected knees after the 12 weeks of intervention between the groups, but not the muscle strength for quadriceps and hamstring muscles, OSI and VAS scores (all *p >* 0.05, Table 2), indicating that the patients on stair-climbing training with IFE have better physical activity than those only on stair-climbing training.

## 4. Discussion

The most important result of this study was the significant difference in WOMAC scores between the patients with knee OA on stair-climbing training with and without IFE, indicating that stair-climbing training with IFE improves physical activity in patients with knee OA than only stair-climbing training. However, all parameters, including quadriceps and hamstring strength, dynamic postural stability, VAS score, and WOMAC score significantly improved in the affected knees after 12 weeks of intervention in each group.

In the present study, knee muscle strength improved after 12 weeks of intervention but did not significantly differ between the two groups. This result may be attributed to the muscle-strengthening exercises performed by both groups. According to a Cochrane systematic review by Fransen et al. [14] therapeutic exercise for knee OA is beneficial in improving knee muscle strength. In particular, several studies have reported that stair-climbing training can effectively improve knee muscle strength [37,38]. Our results support previous findings that stair-climbing training is effective in improving muscle strength. However, IFC therapy may not be directly effective in improving muscle strength in patients with knee OA [39]. Furthermore, in this study, dynamic postural stability improved after 12 weeks of intervention, but was not significantly different between the two groups. Although why no differences were found in dynamic postural stability is unclear, one possible reason may be visual compensation. Postural control can be achieved through reflex mechanisms in the visual, vestibular, and somatosensory systems [40,41]. Li et al. found significant differences between postural control with and without vision compensation in patients with unstable ankles [40]. Similarly, Pirayeh et al. [42] found that eyes closed had worse postural stability than eyes open in patients with knee OA. Thus, visual information plays a significant role in postural control [40,43]. In this study, however, posture was maintained while looking at the screen during the dynamic balance test in both groups. Therefore, dynamic postural stability may not be different between the two groups. Another possible reason may be the balance test method used. O’Connell et al. reported that evaluating postural stability is not appropriate as the ability to control posture while standing on both legs increased significantly [44]. In the present study, for safety reasons in elderly patients, postural stability tests in both groups were performed in a standing position on both legs. Therefore, further prospective studies involving postural stability tests on a single leg without visual compensation are necessary to confirm the results of this study. In addition, Tok et al. reported that although postural stability improved after applying IFC in patients with knee OA, it is unclear whether this improvement is clinically relevant [45]; as exercise training alone can also enhance postural stability in these patients [46].

In the present study, pain significantly improved after 12 weeks of intervention, but was not significantly different between the two groups. However, despite the pain score satisfying the MCID in the stair-climbing training with the IFE group, it did not satisfy the MCID in the stair-climbing training without the IFE group. Although the reason for this is unclear, a possible explanation may be the effect of IFC therapy. According to the results of a systematic review and meta-analysis [22,47] and a narrative review [48], the analgesic effects of IFC therapy can effectively improve pain and physical activity in patients with knee OA. IFC therapy increases the effects on deep tissues by reducing cutaneous sensory nerve stimulation and suppressing the descending pain pathways [48]. Based on the results of our study, we claim that pain recovery in patients with knee OA may be achieved by improving muscle strength and postural stability, but adding IFE with therapeutic exercise may further benefit pain control. Therefore, we aimed to evaluate stair ascent and descent in patients with knee OA to clarify the results of this study. The stair ascent and descent tests are useful for determining knee muscle strength, pain, and physical activity in patients with knee OA [49,50,51].

The WOMAC is the most widely used self-report tool for assessing physical function in patients with knee OA [52]. In this study, the WOMAC score improved after 12 weeks of intervention in each group, but was better in the stair-climbing training with the IFE group than in the stair-climbing training without the IFE group. One possible reason for this result may be the greater degree of quadriceps strength improvement in the stair-climbing training with the IFE group. Kim et al. [53] reported that the WOMAC score significantly correlated with quadriceps strength in patients with knee OA. In this study, quadriceps strength was not significantly different (*p* = 0.052) between stair-climbing training with and without the IFE group; however, a better degree of quadriceps strength improvement was observed in the stair-climbing training with the IFE group than that in the stair-climbing training without the IFE group (46.7% and 28.2%, respectively). In particular, >15% quadriceps strength difference among patients with knee OA after 12 weeks of intervention is clinically important [36]. This improvement is also associated with pain. Bjerre-Bastos et al. [52] reported that pain during weight-bearing activities (such as walking, standing, and climbing stairs) on the WOMAC pain subscale was closely related to the VAS pain score, which correlates with physical activity. Additionally, a systematic review and meta-analysis by Chen et al. [22] found that IFE was effective in alleviating both short- and long-term knee pain and in improving short-term knee function, as measured by the WOMAC score. As mentioned above, in the present study, the VAS pain score also satisfied the MCID in the stair-climbing training with the IFE group, but not in the stair-climbing training without the IFE group.

This study has several limitations. The first limitation is the small sample size and the lack of a normal control group. Therefore, further high-quality studies, including randomized controlled trials, may be required to clarify the results of this study. Second, hip muscle strength was not measured. A systematic review has reported that hip muscle strength plays an important role in function and pain management in patients with knee OA [54]. Additionally, postural stability tests on a single leg without visual compensation and stair ascent and descent tests should be performed. Third, a training period of 12 weeks may be short, and a long follow-up period should be considered to evaluate the long-term effects. Finally, further studies focusing on participants undergoing IFC therapy alone are needed to clarify the results of the present study. Nevertheless, this is the first study to compare knee muscle strength, dynamic postural stability, pain scores, and physical activity in patients with knee OA on stair-climbing training with and without IFE.

## 5. Conclusions

Although knee muscle strength, dynamic postural stability, or pain scores were not different between stair-climbing training with and without IFE in patients with knee OA, stair-climbing training with IFE was more beneficial for physical activity recovery than stair-climbing training without IFE. In particular, stair-climbing training enhanced the recovery of knee muscle strength, dynamic postural stability, pain, and physical activity in knee OA patients with K–L grade ≤ 2. Therefore, clinicians and therapists should be aware that stair-climbing training can be practiced in daily life for the management of patients with knee OA, and the addition of IFE may be considered for better physical activity improvement.

## Figures and Tables

**Figure 1 diagnostics-14-02060-f001:**
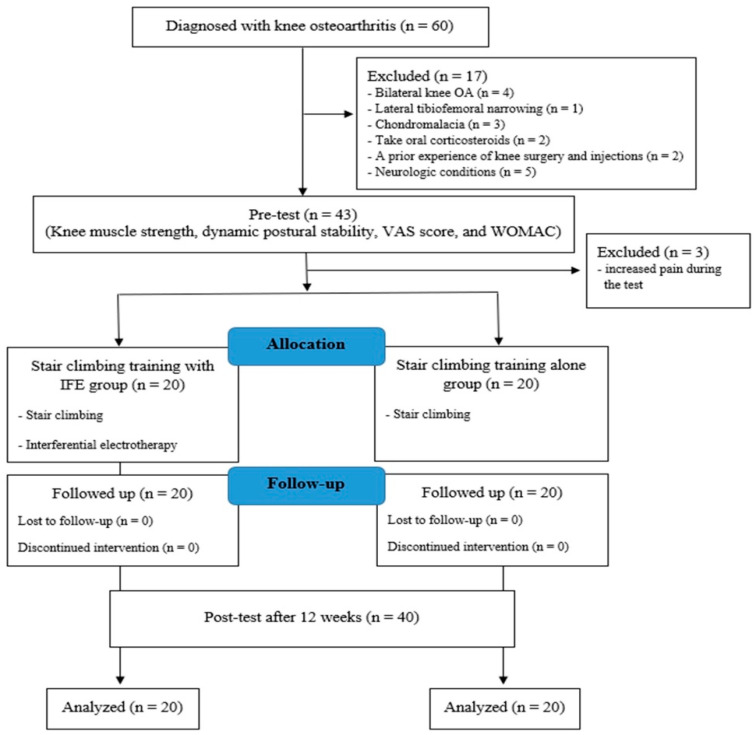
Flowchart of the experimental study.

**Figure 2 diagnostics-14-02060-f002:**
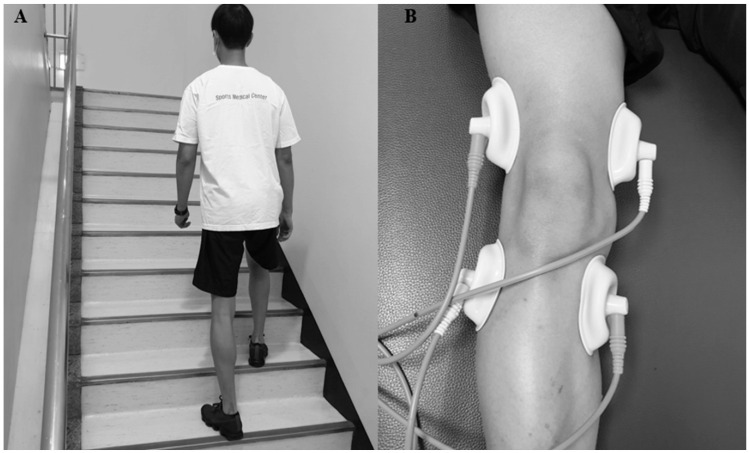
(**A**) Stair climbing training (width, 126.5 cm; depth, 28.5 cm; height, 16.5 cm), (**B**) Interferential electrotherapy (IFE).

**Figure 3 diagnostics-14-02060-f003:**
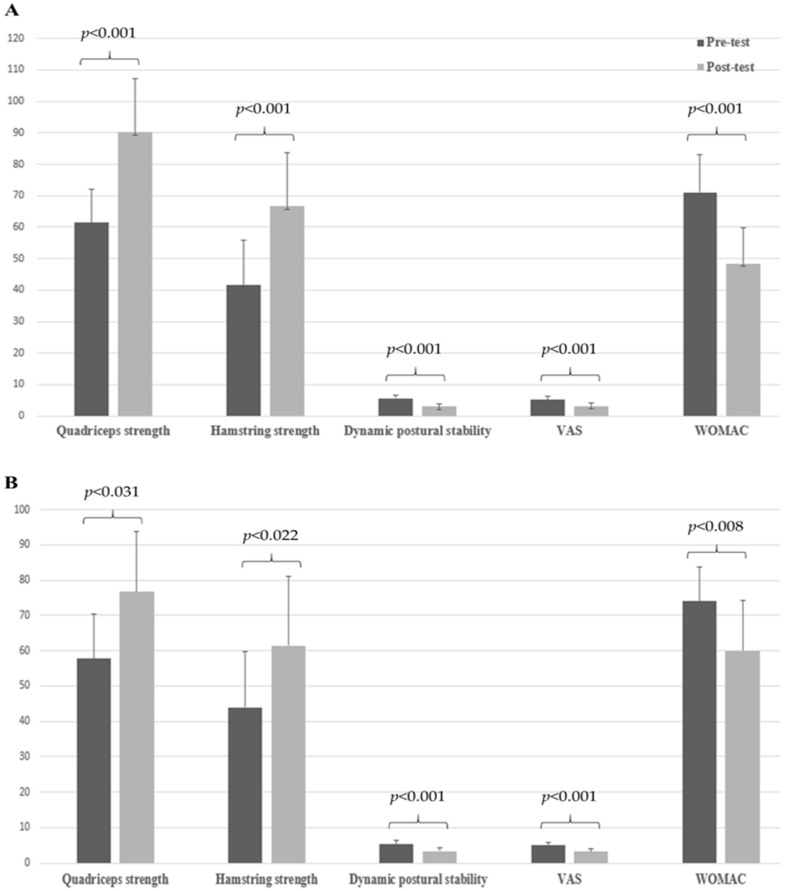
Comparison of knee muscle strength, dynamic postural stability, pain scores, Western Ontario and McMaster Universities Osteoarthritis Index (WOMAC) before and after interventions in each group. (**A**) stair climbing training with interferential electrotherapy (IFE) group, (**B**) stair climbing training without IFE group. Measurement unit of knee muscle strength, dynamic postural stability, VAS, and WOMAC was N·m/kg × 100, degrees, point, and point, respectively.

**Table 1 diagnostics-14-02060-t001:** General characteristics of the participants.

		Stair Climbing Training with IFE (n = 20)	Stair Climbing Training without IFE (n = 20)	*p* Value
Sex	Male	8	10	1.00
	Female	12	10	
Affected knee	Right	16	15	1.00
	Left	4	5	
Age (years) ^a^		65.4 ± 2.8	66.8 ± 1.9	0.348
Height (cm) ^a^		163.4 ± 7.7	166.2 ± 4.1	0.604
Weight (kg) ^a^		64.1 ± 8.3	66.1 ± 4.9	0.462
BMI (kg/m^2^) ^a^		24.1 ± 3.4	23.9 ± 3.1	0.831

IFE, Interferential electrotherapy; BMI, body mass index. ^a^ Values are expressed as mean ± standard deviation.

**Table 2 diagnostics-14-02060-t002:** Comparison of outcome measures between the two groups.

		Stair Climbing Training with IFE (n = 20)	Stair Climbing Training without IFE (n = 20)	*p* Value
Quadriceps strength (N·m/kg × 100)	Pre-test, mean (SD)	61.4 (10.5)	59.8 (12.5)	0.519
	MD, (95% CI)	1.6, (−5.1, 16.8)		
	Cohen’s d	0.14		
	Post-test, mean (SD)	90.1 (17.2)	76.7 (17.1)	0.052
	MD, (95% CI)	13.4, (−16.2, 31.1)		
	Cohen’s d	0.78		
Hamstring strength (N·m/kg × 100)	Pre-test, mean (SD)	41.6 (14.4)	44.0 (15.8)	0.610
	MD, (95% CI)	−2.4, (−8.9, 20.8)		
	Cohen’s d	−0.16		
	Post-test, mean (SD)	66.6 (17.1)	61.4 (19.6)	0.459
	MD, (95% CI)	5.2, (−3.9, 11.3)		
	Cohen’s d	0.28		
Dynamic postural stability (°)	Pre-test, mean (SD)	5.6 (1.0)	5.2 (1.3)	0.310
	MD, (95% CI)	0.4, (−1.2, 0.9)		
	Cohen’s d	0.34		
	Post-test, mean (SD)	3.1 (0.8)	3.3 (1.0)	0.613
	MD, (95% CI)	−0.2, (−1.1, 0.9)		
	Cohen’s d	−0.66		
VAS score (point)	Pre-test, mean (SD)	5.2 (1.0)	5.0 (0.9)	0.716
	MD, (95% CI)	0.2, (−0.5, 0.4)		
	Cohen’s d	0.21		
	Median (25th–75th percentile)	5.0 (3.0–7.0)	5.0 (3.0–7.0)	
	Post-test, mean (SD)	3.2 (0.9)	3.2 (0.7)	0.100
	MD, (95% CI)	0.0, (−0.4, 0.4)		
	Cohen’s d	0		
	Median (25th–75th percentile)	2.5 (1.0–4.0)	3.5 (1.0–5.0)	
WOMAC (point)	Pre-test, mean (SD)	71.0 (12.0)	74.0 (9.6)	0.365
	MD, (95% CI)	−3.0, (−9.6, 7.9)		
	Cohen’s d	−0.28		
	Median (25th–75th percentile)	68.0 (58.8–83.3)	70.0 (56.4–85.6)	
	Post-test, mean (SD)	48.5 (11.3)	60.0 (14.3)	0.019 *
	MD, (95% CI)	−11.5, (−22.3, 4.8)		
	Cohen’s d	−0.89		
	Median (25th–75th percentile)	35.0 (24.0–60.4)	46.0 (30.2–69.7)	

IFE, Interferential electrotherapy; VAS, visual analog scale; WOMAC, Western Ontario and McMaster Universities Osteoarthritis Index; SD, standard deviation; MD, mean difference. * Statistically significant (*p <* 0.05). All data were recorded and described by one physical therapist.

## Data Availability

The data presented in this study are available on request from the corresponding author.

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
