# Peer review of "Stair-Climbing Training with Interferential Electrotherapy Improves Knee Muscle Strength, Dynamic Postural Stability, Pain Score, and Physical Activity in Patients with Knee Osteoarthritis"

_diagnostics, 2024, doi:10.3390/diagnostics14182060_

Round 1
Reviewer 1 Report
Comments and Suggestions for Authors
The authors have done a great job. Despite the positive points, there are a remarks.
Materials and Methods:
Advised to show pipiline to participants. It is unclear how many participants there were. Were they all referred for interferential electrotherapy or not?
Was it a mixed group (stair climbing and electrotherapy)? Was it only an electrotherapy group?
What was the amplitude of the interferential electrotherapy? No information available.
Were stimulation thresholds assessed for the patient?
It is not clear in the methods at which time points measurements were made.
It is unclear what was measured in the "Postural Stability Assessment" section. What is meant by dynamic postural stability?
There are missing units of measurement in the Results section and Figure 3.
Please correct the text, there are several typos in the text, for example:
Line 64: Sixty >60 years of age .... class ≤2 knees ... - I suggest you rephrase the sentence
I can conclude that the methods are not properly described and it is difficult to judge the results. The description of the experiment should be rewritten properly and then the study should be revised.
Reviewer 2 Report
Comments and Suggestions for Authors
Dear authors
The manuscript entitled “Stair-climbing training with interferential electrotherapy improves knee muscle strength, dynamic postural stability, pain score, and physical activity in patients with knee osteoarthritis” aims to compare stair-climbing training with IFE with stair-climbing training without IFE, in improving several functional parameters in patients with Knee OA. This work, despite not configuring an RCT design, has the merit of using a control group. It should also be noted that it is well organized and written in a simple and objective way, which makes it easier to read and understand.
However, given the results obtained, especially the differences between groups obtained in WOMAC, but not in the other indicators, which, as assumed by the authors, would be expected to be related, I think that it deserves a more detailed reflection, specifically in the identification of other possible biases (for example the fact that the groups are not blind). In this sense, I believe that it could add value to the manuscript if there was a reformulation of the discussion regarding the results obtained at WOMAC and reflect these changes in the limitations of the study.
Finally, in table 2, in the VAS post-test MD, the value of (0, ) appears, which must be corrected to (0.0).
Reviewer 3 Report
Comments and Suggestions for Authors
The authors report on the effect of IFC therapy added to stair climbing therapy on pain, strength, stability and activity outcomes. The study is well powered and provides useful information. However, it would have been good to see a group which included IFC therapy alone, since all groups improved with treatment. If IFC has a meaningful impact, it should also improve pain scores without the stair climbing intervention.
Additionally, in the section starting at line 111, it is not clear if the IFC is performed prior to, during or after the stair climbing therapy.
Otherwise, the study is acceptable.
Comments on the Quality of English LanguageThe authors may consider adding the entire section of lines 89-96 before the sentence in line 85 starting with "Forty". This seems more logical than initially describing the final inclusion number and back tracking
Round 2
Reviewer 1 Report
Comments and Suggestions for Authors
The authors have done a great job in improving the manuscript, but there are still some comments.
When discussing, I suggest comparing groups with and without IFE. It will be more objective to discuss relative changes.
I cannot agree with the authors in presenting the results of the VAS and WOMAC scales. VAS are ordinal scale. And I can't figure out what Mean +- STD is for them. Median or mode would be better as a measure of central tendency. Quartiles will be good to measure scatter. In my opinion, median or mode and quartiles would be better for WOMAC. Same remarks for the paired test, t-test is not suitable for rank distributions.
There are some questions for the postural stability test. The duration of time is unclear. Was a stability index or angles used? If stability index was used, the same ranking statistics should be used.
In any case, the way of calculating the stability index or angles should be presented in the Methods.
The Discussion can be supplemented by the following studies:
10.2337/diacare.22.2.328
10.1007/s00421-004-1157-7
10.1109/ITNT55410.2022.9848704
10.1109/CNN56452.2022.9912522
Briefly, the authors write that there are a number of mechanisms for postural stability. And these are well known. The specific mechanism can be determined by spectral analysis and/or a number of specific tests. To improve outcomes, patients with proprioceptive stability control should be distinguished from all and compared separately.
Author Response
"Please see the attachment."
